# *DRL1*, Encoding A NAC Transcription Factor, Is Involved in Leaf Senescence in Grapevine

**DOI:** 10.3390/ijms20112678

**Published:** 2019-05-31

**Authors:** Ziguo Zhu, Guirong Li, Chaohui Yan, Li Liu, Qingtian Zhang, Zhen Han, Bo Li

**Affiliations:** 1Shandong Institute of Pomology, Shandong Academy of Agricultural Science, No 66 Longtan Road, Taian 271000, China; shanhong98@163.com (Z.Z.); 15153871569@163.com (L.L.); tcszqt@163.com (Q.Z.); hanzhen86@163.com (Z.H.); 2College of Horticulture and Landscape Architecture, Henan Institute of Science and Technology, Xinxiang 453003, China; liguirong10@163.com (G.L.); ychputao@163.com (C.Y.)

**Keywords:** Grapevine, NAC transcription factor, ABA, leaf senescence

## Abstract

The NAC (for NAM, ATAF1,2, and CUC2) proteins family are plant-specific transcription factors, which play important roles in leaf development and response to environmental stresses. In this study, an NAC gene, *DRL1*, isolated from grapevine *Vitis vinifera* L. “Yatomi Rose”, was shown to be involved in leaf senescence. The quantity of *DRL1* transcripts decreased with advancing leaf senescence in grapevine. Overexpressing the *DRL1* gene in tobacco plants significantly delayed leaf senescence with respect to chlorophyll concentration, potential quantum efficiency of photosystem II (Fv/Fm), and ion leakage. Moreover, exogenous abscisic acid (ABA) markedly reduced the expression of *DRL1*, and the ABA and salicylic acid (SA) concentration was lower in the *DRL1*-overexpressing transgenic plants than in the wild-type plants. The *DRL1* transgenic plants exhibited reduced sensitivity to ABA-induced senescence but no significant change in the sensitivity to jasmonic acid-, SA- or ethylene-induced senescence. Transcriptomic analysis and RNA expression studies also indicated that the transcript abundance of genes associated with ABA biosynthesis and regulation, including 9-*cis*-epoxycarotenoid dioxygenase (*NCED1*), *NCED5*, zeaxanthin epoxidase1 (*ZEP1*), ABA DEFICIENT2 (*ABA2*), *ABA4*, and ABA INSENSITIVE 2 (*ABI2*), was markedly reduced in the *DRL1*-overexpressing plants. These results suggested that *DRL1* plays a role as a negative regulator of leaf senescence by regulating ABA synthesis.

## 1. Introduction

Aging is one of the most important physiological phenomena in biology, and in plants, the leaf is one of the organs most sensitive to senescence. Leaf senescence is not a passive degradation process, but a developmentally programmed process, regulated in an orderly fashion [1,2]. During leaf senescence, there are considerable changes in cell structure, hormone level, physiology, biochemical metabolism, and gene regulation [1].

Plant hormones play an important regulatory role in leaf senescence [1]. Cytokinins can inhibit leaf senescence and prolong leaf life while overexpression of isopentenyl-transferase (*IPT*), the key gene for cytokinin synthesis, can delay plant senescence [3,4]. Abscisic acid (ABA) can promote plant senescence with exogenous ABA application inducing the expression of senescence-associated genes (SAGs) and ABA-synthesis genes such as *NCED2*, *NCED3*, *AAO1*, and *AAO3* in *Arabidopsis thaliana* (L.) Heynh and can lead to premature leaf senescence [5,6,7]. Ethylene and jasmonic acid (JA) can also promote plant senescence, and genes involved with the biosynthesis of ethylene (ET) and jasmonic acid (JA) were up-regulated following the exogenous application of the respective hormone [8,9]. Both the ethylene-insensitive mutants *etr1* and *ein2* and the JA-insensitive mutants *coi1* of *A. thaliana* showed a delayed-aging phenotype [10,11,12]. Salicylic acid (SA) plays a role in promoting natural leaf senescence with *A. thaliana* plants defective in the SA-signaling pathway (namely *npr1* and *pad4* mutants) exhibiting reduced expression of SAGs and delayed-senescence phenotypes [13]. However, the effects of auxins or gibberellins (GA) on leaf senescence are not clear.

Leaf senescence is a complex developmental phase. The onset and progression of leaf senescence are controlled by an array of signal transduction genes [2]. NAC (NAM, ATAF1,2, and CUC2)-domain proteins are plant-specific transcription factors [14,15,16]. In the genomes of *A. thaliana*, *Populus trichocarpa* L., *Glycine max* L., and *Oryza sativa* L., 117, 120, 152, and 151 NAC genes have been identified [17,18,19,20], respectively. NAC proteins contain a highly conserved N-terminal domain that is responsible for DNA binding activity and a variable C-terminal domain that ensures the specificity of NAC transactivation activity [15]. Many members of the NAC transcription factor family have been reported to be functionally involved in the regulation of leaf senescence in *A. thaliana* [21,22], wheat [23], bamboo [24], and rice [25]. For example, the *A. thaliana* mutant (*anac092*/*atanc2*/*ore1*) exhibited delayed leaf senescence [21]. *AtNAP* is strongly up-regulated during leaf senescence in *A. thaliana*, and *atnap* null mutants show a delayed leaf senescence phenotype, whereas the inducible overexpression of *AtNAP* causes precocious leaf senescence [22]. In wheat, overexpressing *TaNAC-S* resulted in delayed leaf senescence associated with increased grain yield and protein concentrations [23], whereas in bamboo, constitutive overexpression of *BeNAC1* resulted in precocious senescence phenotypes in *A. thaliana* [24], and overexpression of *OsNAC2* dramatically accelerated leaf senescence in *Oryza sativa* [25].

Grapevine (*Vitis vinifera* L.) is an important fruit crop cultivated worldwide [26]. A total of 74 NAC genes were identified from the grapevine genome [27]. Several members of NAC in grapevine have been found to participate in development, abiotic, and biotic stress. Over-expression of *VvNAC1* in *A. thaliana* enhanced tolerance to osmotic, salt, and cold stresses and to *Botrytis cinerea* and *Hyaloperonospora arabidopsidis* pathogens [28]. *NAC26* from *V. Amurensis* showed the tolerance to drought stress by the regulation of JA in *A. thaliana* [29]. A different *VvNAC26* polymorphism and their combination were involved in berry size variation in grapevine [30]. However, the detailed function of grapevine NAC genes in terms of their role in leaf senescence is unknown. In our previous study, a NAC transcription factor gene *DRL1* (GenBank No. XP-002281816) was identified from grapevine which was associated with dwarf plant and rolled leaf phenotypes [31]. In the present study, we report that *DRL1* was involved in leaf senescence, and the regulatory mechanism might be associated to the ABA signaling pathway.

## 2. Results

### 2.1. DRL1 is Down-Regulated during Leaf Senescence in Grapevine

Leaf senescence seriously affects photosynthesis and nutrient assimilation, and subsequently influences the yield and quality of grapes [32,33]. To confirm the role of the *DRL1* gene in leaf development, we determined its transcript abundance at different stages of leaf development in grapevine. The expression level of the *DRL1* gene was highest in the young leaf, decreasing with leaf age, and reaching the lowest level in leaves at late senescence (Figure 1a,b). In other tissues, including root, stem flower, and fruit, the transcripts of *DRL1* kept to a low level (Figure 1b). Transgenic tobacco plants were generated, which expressed the GUS protein driven by the promoter of the *DRL1* gene. A histochemical staining analysis showed that the *DRL1* gene was mainly expressed in the stem and leaves (Figure 1c). In 14-day-old transgenic plants, more GUS protein was detected in the young leaf, which showed dark blue, whereas less GUS protein was detected in the old leaf, which showed only slightly blue. A similar result was obtained in 4-week-old transgenic plants (Figure 1d). These results show that the *DRL1* gene may be involved in leaf senescence.

### 2.2. Ectopic Overexpression of DRL1 Delayed Leaf Senescence

We confirmed the role of *DRL1* in leaf senescence by monitoring leaf color as a marker of senescence. Wild-type (WT) tobacco plants exhibited the visible yellowing phenotype in older leaves compared with the leaf at the same stage of development in similar-aged plants of the *DRL1*-overexpressing transgenic plants (Figure 2a). In 10-week-old plants, the survival rate of leaves in the transgenic plants was higher than that in WT plants, the survival rate being 55.6% in transgenic plants, while only 39.1% in WT tobacco plants (Figure 2b, Appendix A). In 12-week-old plants, there was still a 30.2% survival rate of leaves in the transgenic plants, but no leaves (green or yellow) remained on the WT plants (Figure 2b). Further physiological analyses of leaf senescence were performed on 8-week-old tobacco plants. The delay in leaf senescence in the transgenic plants compared to their WT counterparts was also supported by changes in total chlorophyll (Chl) concentration (Figure 2c), ion leakage (Figure 2e), and the potential quantum efficiency of photosystem II in the dark-adapted state (*Fv*/*Fm*) (Figure 2d). Moreover, DRL1 transgenic plants exhibited a delay in the plant height and the leaves curled inward compared with the wild type. To eliminate potential indirect effects of growth retardation on leaf senescence, the 12^th^–14^th^ leaves of plants that exhibited similar patterns of leaf aging were assayed. After treatment in dark for 5 d, WT plant leaves retained 65% of their chlorophyll concentration, while transgenic plants retained 82% of their Chl concentration. Hence, the retardation and curled leaves did not affect the function of leaf senescence (Appendix A). These results suggest that *DRL1* negatively regulates leaf senescence.

### 2.3. Analysis of Differentially Expressed Genes (DEGs) in DRL1-Overexpressing Transgenic Plants

To investigate the possible regulatory roles of *DRL1* at the transcriptional level, we generated cDNA libraries from leaves of *DRL1* transgenic and WT tobacco plants and performed the genome-wide expression profile analysis using the Illumina platform (Appendix A). In the whole unigene set, a total of 44265 unigenes were significantly matched to known genes in the public databases of NR, NT, Swiss-Prot, KEGG, COG, and GO (Appendix A). A total of 1149 differentially expressed genes (DEGs) were screened out using the combined criteria of at least a two-fold change and a significant chi-square test (*p* < 0.05, false detection rate (*FDR*) < 0.01). Among these DEGs, 584 were up-regulated and 565 were down-regulated in the transgenic plant relative to the WT (Figure 3a, Appendix A). According to GO annotation and category, these DEGs were involved in regulating gene, development, transport, stress, carbohydrate metabolism, degradation, and secondary metabolism, and unknown gene for the biological process categories, of which the largest number of sequences were those associated with regulatory gene pathways (except for unknown genes) (Figure 3b).

To further characterize the regulatory genes, DEGs of the regulatory gene category were further subdivided into seven subgroups, including hormone, transcription, kinase, protein-binding, signaling, phosphatase, and calcium. The largest number of regulatory sequences was associated with hormones (Figure 3c, Appendix A). Among these hormone-related genes, the largest number of genes were those related to auxin (IAA) and ABA. The IAA-related genes included 13 up-regulated genes and 19 down-regulated genes, while the ABA-related genes contained two up-regulated genes and ten down-regulated genes (Figure 3d). We speculated that IAA and ABA play a key role in leaf senescence in the *DRL1*-overexpressing transgenic tobacco plants.

### 2.4. ABA Is Involved in Leaf Senescence of DRL1-Overexpressing Transgenic Plants

To further analyze the role of the hormone ABA in leaf senescence, we investigated the hormone concentration in the transgenic and WT tobacco plants. The results are shown in Figure 4. Among the six major plant hormones, there was no significant change in auxins (IAA, IBA, ICA, ME-IAA), cytokinins (cZ, TZ, DZ, IP), gibberellins, or jasmonic acid between *DRL1*-overexpressing transgenic and WT plants. However, the concentrations of salicylic acid (SA) and ABA in the transgenic plants were significantly lower, with reductions of 32.8% and 75.4%, respectively, compared with WT plants.

To test whether *DRL1* is involved in ABA-induced leaf senescence, we compared the senescence response of 8-week-old leaves of both transgenic and WT plants following exogenous ABA treatment. By 9 d after ABA treatment, WT leaves had lost most of their Chl concentration (Figure 5a). In contrast, the leaves of the transgenic plants had retained 30% of their Chl concentration (Figure 5b). We then carried out the same experiment using detached leaf disks (Figure 5c). After 7 d of ABA treatment, WT leaf disks retained 50% of the Chl concentration retained by the disk of the transgenic plants. In addition, we tested whether *DRL1* was also involved in the leaf senescence response to other hormones (Figure 5d). After 9 d of exogenous MeJA, SA, or ETH treatments, there was no significant difference in leaf color between transgenic and WT plants, both of which had lost most of their Chl concentrations and only retained 11% of their Chl concentration. Moreover, IAA, GA, and CTK did not affect leaf senescence by the regulation of *DRL1* (Appendix A).

### 2.5. DRL1 Delayed Leaf Senescence by Regulation of ABA

To better understand the regulatory mechanism of *DRL1* in the process of leaf senescence, we investigated the expression profile of *DRL1* in response to exogenous application of the hormone ABA in grapevine. The expression of *DRL1* decreased following treatment with ABA. The transcripts showed a strong reduction of 99.9% at 48 h after treatment with ABA compared with the control (Figure 6a). The promoter-inducible activity was assessed using GUS histochemical staining in transgenic plants. Following treatment with water (Control check, CK), 7-day-old transgenic seedlings were heavily stained blue, whereas in plants treated with ABA, only the leaf margins were stained blue, and they were only lightly stained (Figure 6b). To understand how *DRL1* controls leaf senescence at the molecular level, we examined the genes related to ABA biosynthesis and regulation (Figure 6c). The ABA biosynthesis genes were NCED and *ZEP1*. The expressions of *NCED1*, *NCED2*, *NCED3*, *NCED5*, and *ZEP1* were significantly down-regulated in transgenic tobacco leaves compared with the WT. In particular, *NCED3* and *NCED5* transcripts in transgenic tobacco showed remarkable reductions, 95.1% and 93.8%, respectively, compared with WT. With respect to the ABA regulatory genes, the transcripts of *ABA2, ABA4, ABI2*, and *PYL4* were also suppressed in the transgenic plants, with 75.7%, 60.3%, 43.7%, and 89.9% reductions, respectively, compared with the WT. These results indicated that ABAs play an important role in leaf senescence.

## 3. Discussion

Leaf senescence is a complex developmental process, the regulation of which involves many highly organized molecular and cellular processes [1,36]. The *A. thaliana* transcriptome showed that about 10% of the genes were differentially expressed during the late developmental period of leaf senescence, including 134 genes encoding transcription factors such as NAC, WRKY, AP2/EREBP, and MYB [37,38,39]. Of the transcription factors, the NAC family is particularly rich in senescence-regulated transcription factors in many plant species [40,41]. In *A. thaliana*, approximately 20 NAC genes have been shown to exhibit increased expression in senescing leaves [42], while in *Gossypium hirsutum* L., the NAC proteins GhNAC11, GhNAC20, GhNAC57, and GhNAC78 undergo expression changes during leaf senescence [43]. In our study, RT-qPCR showed the opposite expression pattern of *DRL1*, which was down-regulated during natural leaf senescence. The result of histochemical staining was also consistent with the RT-qPCR data, with *DRL1* being mainly expressed in adult leaves and a lower level in old and yellowing leaves, indicating that *DRL1* might be involved in a negative manner in leaf senescence.

Most NAC genes related to leaf senescence play a positive role, such as *AtNAP* [22], *NAM-B1* [44], *NTL4* [45], and *ANAC092/AtNAC2/ORE1* [46]. A few studies have concerned a negative role for NAC genes in leaf senescence. The *A. thaliana* NAC transcription factor *JUNGBRUNNEN1* and *VNI2* negatively regulate leaf senescence [47,48]. In wheat, overexpression of *TaNAC-S* resulted in delayed leaf senescence [23]. To better understand the role of *DRL1* in leaf senescence, physiological analyses of the *DRL1*-overexpressing transgenic plants were performed. Tobacco is easily transformed using Agrobacterium and usually as a model plant organism for studying fundamental biological processes [49,50]; hence, we choose tobacco for the functional analysis of *DRL1*. Compared with the WT tobacco plants, leaves exhibiting over-expression of *DRL1* displayed significantly increased Chl concentration, markedly increased Fv/Fm ratios, and reduced levels of ion leakage. Chl concentration is an indicator of plant nutrition, photosynthesis ability, and developmental stage, while the Fv/Fm ratio is an indicator of the photochemical quantum efficiency of PSII [51]. We predicted that *DRL1* might act on leaf senescence as a negative regulator.

Leaf senescence is regulated by both internal and external factors. Internal factors include leaf age, developmental stage, and endogenous hormone levels, which, in turn, are controlled by regulatory genes [52]. Transcriptome analysis of *DRL1*-overexpressing transgenic plants showed that the largest number of DEGs (except for the unknown gene) corresponded to the regulatory genes (24.6%), of which 26.8% were related to hormones. Plant hormones play a key role in plant growth and development, as well as the response to external stimuli. Comparing the hormone concentrations in transgenic and WT tobacco plants, there was no significant change in auxin, cytokinin, gibberellin, or jasmonic acid concentrations, but concentrations of both SA and ABA were significantly reduced in the transgenic plants. Previous studies reported that SA, ET, and ABA can promote leaf senescence. In *A. thaliana*, overexpression of the transcription factor gene *WRKY75* accelerated leaf senescence by up-regulating the genes related to the SA signal pathway [53]. The *A. thaliana* mitochondrial protease *FtSH4* induced leaf senescence via up-regulation of the WRKY-dependent salicylic acid signal pathway [54]. To determine whether reduced SA and ABA concentrations were the only reason behind the delayed leaf senescence of *DRL1*-overexpressing transgenic plants, we investigated the senescence process in transgenic leaves treated with different hormones. The results supported the hypothesis that ABA plays a key role in *DRL1*-regulated leaf senescence, as the ABA-treated plants retained more Chl than those treated with exogenous SA, ET, and MeJA. Like *A. thaliana* NAC transcription factor *VNI2* [48], grapevine *DRL1* delayed leaf senescence in tobacco by mediating ABA signals.

It is well known that ABA participates in leaf senescence. Some SAGs are induced by ABA and then promote plant senescence, including *SAG113* [55], *AtNAP* [22], and *VNI2* [48]. *DRL1* showed an expression pattern opposite to that of SAGs, with a marked decrease in response to ABA application, and the results were also obtained from *DRL1* promoter analysis. The expression of *RAV1* was also repressed by ABA, but constitutive expression in *A. thaliana* caused premature leaf senescence [56]. Endogenous ABA concentration increases significantly in many plants during leaf senescence. In *Solanum lycopersicum* L., the ABA concentration increased linearly with time during leaf senescence [57]. In *Zea mays* L., the ABA concentration in “stay-green” phenotype varieties was significantly higher than that in premature-senescence varieties, which may be the reason for the delayed senescence in “stay-green” varieties [51]. These results indicated that regulation of ABA biosynthesis plays a key role in leaf senescence.

In the present study, expression analyses also confirmed that overexpression of *DRL1* caused drastic down-regulation of ABA biosynthetic genes, with abundance of *NCED1*, *NCED2*, *NCED3*, *NCED5* and *ZEP1* transcripts all being notably suppressed. NCEDs are the key enzymes controlling ABA production, being involved in xanthophyll cleavage [58]. ZEP catalyzes the conversion of zeaxanthin into violaxanthin in ABA biosynthesis [59]. Ectopic expression of the rice NAC transcription factor gene *OsNAC2* led to an increase in ABA levels via directly upregulating the expression of the ABA biosynthetic genes *NCED3* and *ZEP1* [60]. *SlNAP2* promotes leaf senescence in tomato by directly controlling the expression of genes for ABA biosynthesis, such as genes encoding *NCED1*, ABC transporter G family member 40, and ABA 8′-hydroxylase [61]. Moreover, the transcript levels of the regulatory genes, *ABA2*, *ABA4*, and *ABI* were also sharply reduced in *DRL1*-overexpressing transgenic tobacco plants. ABA2 encodes a short-chain dehydrogenase/reductase1 that catalyzes the multi-step conversion of xanthoxinto abscisic aldehyde during ABA biosynthesis in *A. thaliana* [62]. Transgenic plants overexpressing *AtABA2* have been shown to increase ABA concentration, promote seed dormancy, and increase stress tolerance [63]. The *A. thaliana* mutant aba4 showed reduced endogenous ABA concentration in rosettes and seeds [64], while the *A. thaliana* mutant *abi2* showed higher ABA concentration in plants and sensitivity to ABA [65]. Therefore, we speculated that *DRL1* might delay leaf senescence by down-regulation of ABA biosynthesis.

## 4. Materials and Methods

### 4.1. Plant Material and Hormone Treatment

*V. vinifera* cv. “Yatomi Rose” was grown in the grape germplasm resource orchard of Shandong Institute of Pomology, Taian, Shandong, China. Leaves at different stages of development were collected from the upper, middle, and lower part of the branch. One-year-old rooted cuttings of *V. vinifera* cv. Yatomi Rose were grown in a greenhouse for hormone treatments. The grapevine cuttings were sprayed with 100 µM abscisic acid (ABA) containing 0.05% Tween 20 [66]. Leaves sprayed with 0.05% Tween 20 solution were used as a negative control. Leaves were harvested at 0, 3, 6, 9, 12, 24, 36, and 48 h after treatment with ABA, and immediately frozen in liquid nitrogen. Leaves from at least three replicate plants were collected and pooled to constitute one replicate for each time point, with three biological replicates for each treatment. To examine hormone-induced leaf senescence, the 6-week-old tobacco leaves were floated on 3 mM 2-Morpholinoethanesulfonic Acid (MES) buffer (pH 5.8) containing 50 mM ABA, 50 µM SA, 50mM MeJA, or 5 mM ethylene. All chemical treatments were performed at 22 °C under continuous lighting. All stress-induction experiments were performed on three independent replicate plants for each treatment.

### 4.2. RNA Analysis

Total RNA was extracted using the method described by Zhang et al. [67], and first-strand cDNA was synthesized from 2 µg of total RNA at 42 °C with PrimeScript RT Reagent Kit (Takara, Dalian, China). All PCRs were performed on a Bio-Rad IQ-5 thermocycler (Bio-Rad Laboratories, Berkeley, CA, USA) with 40 cycles of 5 s at 94 °C and 30 s at 58 °C. Cycle threshold values were determined by IQ-5 Bio-Rad software assuming 100% primer efficiency. All primers are listed in Table 1. Three mRNA samples from three independently harvested replicate leaf samples were qPCR analyzed. The relative mRNA ratios were calculated as 2^−ΔΔC*T*^ [68]. The transcript levels of target genes were normalized against *VvActin* for the grape samples and *NtActin* for *Nicotiana* samples [69,70].

### 4.3. Production of Transgenic Tobacco

DNA from the grapevine leaves was extracted using the CTAB (Cetyltrimethylammonium bromide) method. The DRL1 promoter was amplified using the PrimerSTAR GXL DNA Polymerase (Takara, Dalian, China) with the forward primer 5′-GATGTTTGAAATATTTTTAAA TAAGTA-3′ and reverse primer 5′-GTCTACTGCTTACTGGTGTTGTCCTGTGA-3′. The PCR reaction had the following thermal profile: 94 °C for 3 min, 28 cycles of 94 °C for 30 s, 60 °C for 30 s, and 72 °C for 60 s, followed by 72 °C for 10 min. The 1140-bp promoter of *DRL1* was inserted into the vector pCAMIA1391z to generate the construct *DRL1_pro_:GUS*. The transformation of tobacco was performed as described by Horsch et al. [49]. The selection of the transformed plants was conducted by culturing the plantlets on MS medium containing hygromycin (25 mg L^−1^) under 14 h of light and 10 h of dark. The T_1_ generation of transgenic tobacco was used for further analysis.

### 4.4. Chlorophyll Concentration, Fluorescence, and Ion Leakage Analysis

The tobacco plants were placed in a plant culture room at 25–28 °C, 14 h light/10 h dark, 2000 Lx (Illumination intensity). Chlorophyll of 0.5 g leaves was extracted using 80% acetone and analyzed using a UV-Visible Spectrophotometer (UV2600, Shimadzu, Kyoto, Japan) at 645 and 653 nm [70]. Relative electrolyte leakage was assessed by ion leakage analysis as previously described by Zhang [57]. Potential quantum efficiency for PSII (F*v*/F*m*) was measured as described by Jiang [71].

### 4.5. Transcriptome Analysis

Transgenic tobacco plants and the WT plants were divided into two groups, each with two independent biological replicates. Mature leaves collected from the middle of the tobacco plants were used for total RNA isolation. The differentially expressed gene (DEG) experiments were performed by the Biomarker Technology Company (Beijing, China). Total reads were generated using the HisSeq^TM^2500. The raw data were first purified by trimming adapters and removing low quality sequencing. All adaptor-trimmed reads were mapped to the tobacco genome (ftp://ftp.solgenomics.net/ genomes/Nicotiana-tabacum/) using a Basic Local Alignment Search Tool (BLAST)-like alignment tool (BLAT). For the differentially expressed genes assay, FPKM (fragments per kilobase per million reads) values were used to normalize gene expression levels [72]. DEGS between the transgenic and WT plants were screened using DESeq software [73], based on a false discovery rate (FDR) < 0.01 and |log_2_ (fold change)| >1 or <−1. For the annotation and categorization, the DEGs were subjected to the GO database (http://www.geneontology.org/) and the Kyoto Encyclo- pedia of Genes and Genomes (KEGG) database (http://www.genome.jp/kegg/pathway.html) using Blast2GO and KEGG Automatic Annotation Server (KAAS, http://www.genome.jp/kegg/kaas/), respectively. For the categorization, the DEGs were analyzed for GO category enrichment and KEGG pathway enrichment using GO-seq and KAAS, respectively [74].

### 4.6. Hormones Analysis

For transgenic and WT tobacco plants, leaves in the middle of the plant from 6-week-old OE1 (*DRL1*-overexpressing plants) and WT were harvested under normal conditions. The hormones were extracted and quantified by MetWare (http://www.metware.cn/) based on a LC-MS/MS platform (MetWare Company, Wuhan, China). Three biological replicates of each assay were performed.

## 5. Conclusions

Our study demonstrated that *DRL1* gene from grapevine plays a negative role in leaf senescence. Overexpression of *DRL1* gene in tobacco plants significantly delayed leaf senescence. Moreover, transcriptomics and hormone analysis indicated that the transcript abundance of genes were associated with ABA biosynthesis. The exogenous hormones treatment to transgenic plants and RNA expression analysis further confirmed that *DRL1* delayed leaf senescence by regulation of ABA signal pathway. But which gene is the direct target gene of the *DRL1*, *NECD*, *ZEP*, *ABA*, *ABI*, or other genes? This will be our future research which will be helpful to know the regulation mechanism of leaf senescence in grapevine.

## Figures and Tables

**Figure 1 ijms-20-02678-f001:**
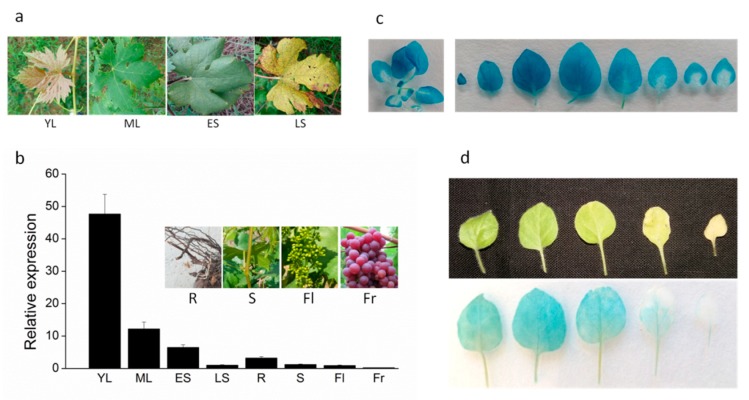
The expression of *DRL1* during leaf senescence in grapevine. (**a**) Leaf senescence phenotypes in grapevine cultivar ‘Yatomi Rosa’. YL, expanding young leaf; ML, mature leaf that is fully expanded but with no visible yellowing; ES, early senescence leaf, with about 5% of the leaf area yellowing; LS, late senescence leaf, with more than 50% of the leaf area yellowing. (**b**) Relative expression of *DRL1* during leaf senescence. The expression levels were determined using qPCR. The expression levels in LS leaves were set to 1. Bars represent mean ± standard deviation (SD). (**c**, **d**) GUS staining in leaves of P_DRL1_-GUS transgenic tobacco plants at 10 d (**c**) and 20 d (**d**).

**Figure 2 ijms-20-02678-f002:**
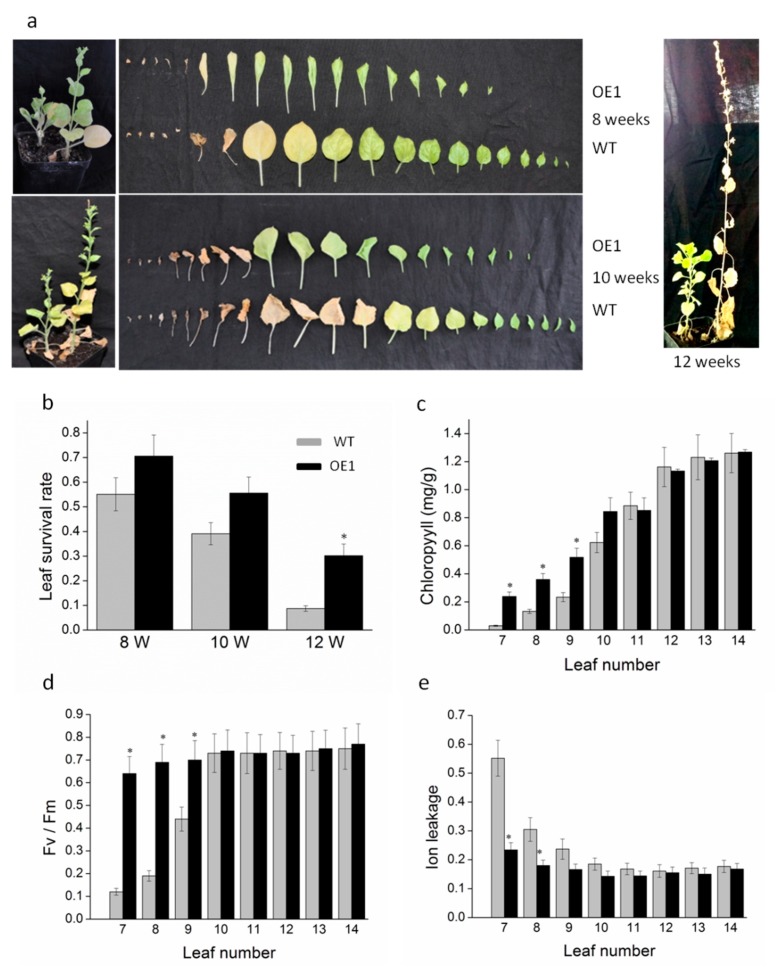
DRL1-over-expression delayed leaf senescence. (**a**) The phenotype of transgenic and wild-type (WT) tobacco plants at 8, 10, and 12 weeks. (**b**–**e**) Physiological analyses of leaves from transgenic plants. Leaf survival rate (**b**), chlorophyll concentration (**c**), ion leakage (**d**), and *Fv*/*Fm* ratio (**e**) were measured in WT and transgenic plants of the same age. Bars represent mean ± SD. Significant difference from the WT was confirmed by Tukey’s test [34,35] (* *p* < 0.05).

**Figure 3 ijms-20-02678-f003:**
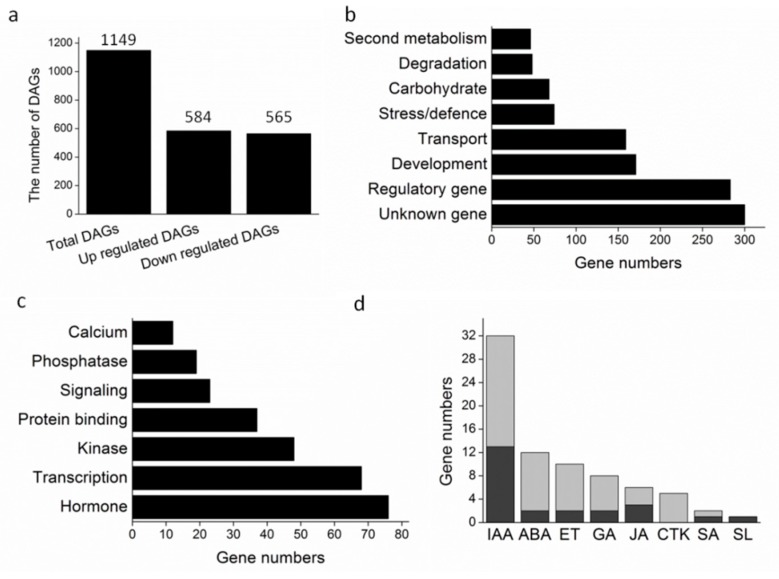
DRL1 regulates various genes associated with leaf senescence. (**a**) The number of DEGs in DRL1-overexpressing transgenic plants compared with wild type plants. (**b**) Classification of DAGs based on Gene Ontology (GO) annotation. (**c**) Classification of regulatory genes. (**d**) Classification of hormone-related genes (gray columns represent up-regulated genes, black columns represent down-regulated genes).

**Figure 4 ijms-20-02678-f004:**
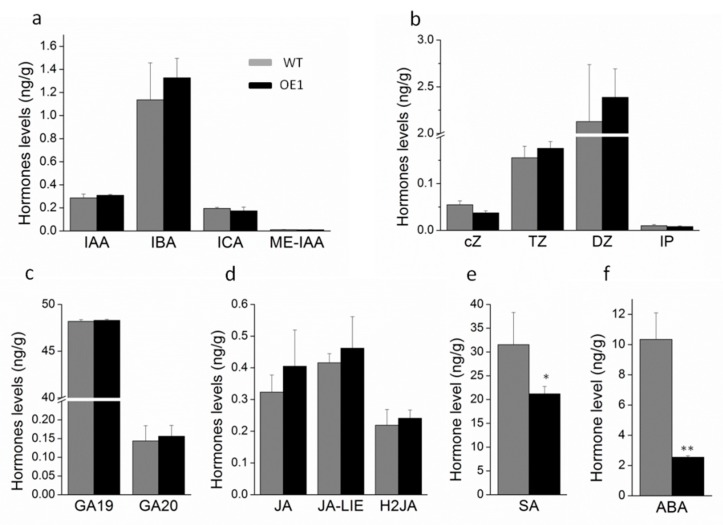
The concentrations of the hormones auxins (**a**), cytokinins (**b**), gibberellins (**c**), jasmonic acid (**d**), salicylic acid (**e**), and abscisic acid (**f**) in DRL-overexpressing transgenic and wild-type tobacco leaves. Bars represent mean ± SD. Significant difference from the WT was confirmed by Tukey’s test *(** *p* < 0.05 and ** *p* < 0.01).

**Figure 5 ijms-20-02678-f005:**
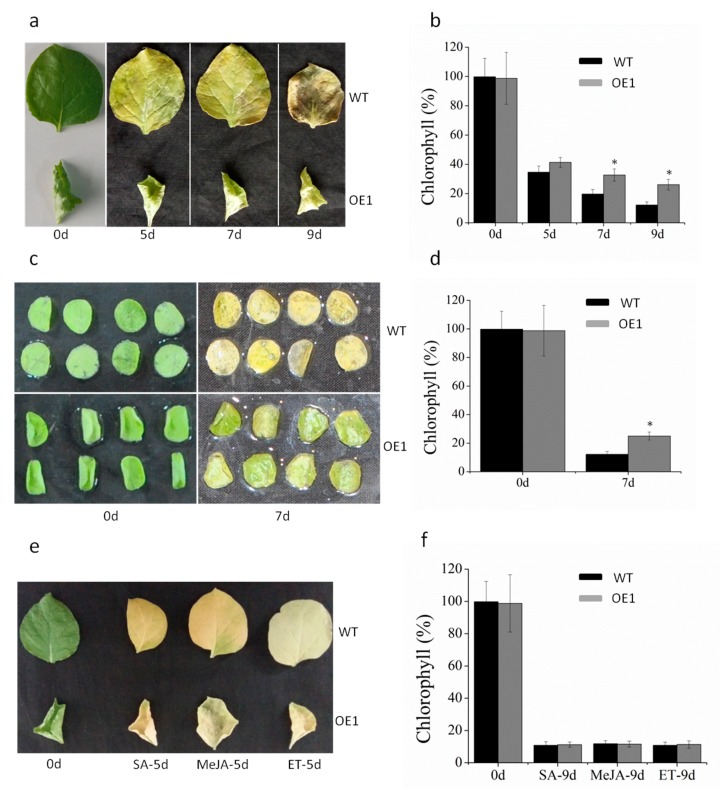
Leaf senescence induction by application of hormones to *DRL1*-overexpressing transgenic tobacco plants. (**a**, **b**) Detached leaf senescence phenotypes of transgenic tobacco treated with abscisic acid (**a**) and determination of chlorophyll concentration (**b**). (**c**, **d**) Leaf senescence phenotypes of transgenic tobacco leaf disks treated with ABA (**c**) and determination of chlorophyll content (**d**). (**e**) Leaf senescence phenotypes of transgenic tobacco treated with salicylic acid, methyl jasmonate, or ethylene followed by determination of chlorophyll content (**f**). Bars represent mean ± SD with at least three biological replicates. Significant difference from the WT was confirmed by Tukey’s test *(** *p* < 0.05).

**Figure 6 ijms-20-02678-f006:**
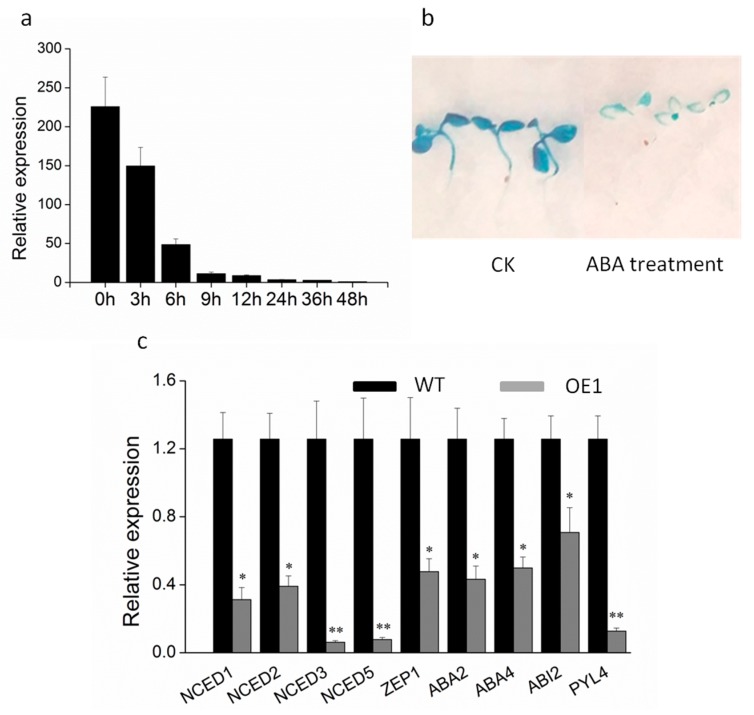
The expression of abscisic acid metabolism-related genes in WT and *DRL1*-overexpressing transgenic plants. (**a**) The expression of *DRL1* in grapevine following ABA treatment. (**b**) GUS staining in leaves of P_DRL1_-GUS transgenic tobacco plants under ABA treatment. Six-day-old tobacco seedlings were treated (*n* = 60). (**c**) Expression analysis of ABA biosynthesis- and regulation-related genes in wild-type and *DRL1*-overexpressing transgenic tobacco plants. Asterisks represent statistically significant differences between wild-type and transgenic plants: * *p* < 0.05 and ** *p* < 0.01.

**Table 1 ijms-20-02678-t001:** Primers in the experiment for PCR.

Gene Name	Forward Primer 5′→3′	Reverse Primer 5′→3′
*NtNCED1* (NM_001325669)	gtttggtggggagcctctgttttta	tgatttccattctttctcattgtgc
*NtNCED2* (NM_001326185)	tccacttcaaaaccaaccactatt	ttaaggcactttccacggcatcta
*NtNCED3* (JX101473)	ctcacacgaactccaaaacccactt	tagcaccattacggacataaacccc
*NtNCED5* (XM_016625112)	ctccaaaacccacttcctaaaaca	agcaccattacggacataaacccc
*NtZEP1* (XM_016582459)	tagaggaccaattcagatacagagca	agttaccagaaacaccatcaaccaag
*NtABA2* (EU123520)	ggtgctgatggcataaggtctaa	ctccacccacatctgaagaaaca
*NtABA4* (XM_016637325)	cttccacttgcttttgtcactccc	cacccatttctatgcttacttccc
*NtABI2* (NM_001324946)	tgttatgcagggtggtgttaaaggc	aaggtgaaggtgaaagagatgatgt
*NtPYL4* (XP_00977852)	atcggtctgttactactttacatcct	cctttcgtctacccaaattctca
*NtActin* (XM_016618073)	tggagaaaatctggcatcacacg	actggcataaagggatagaacgg
*VvDRL1* (XP-002281816)	ccgggtttcggtttcatccta	actccctctcgccaatcttcg
*VvActin* (AF369524)	cctcaaccccaaggccaacaga	accatcaccagaatccagcaca

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
