# Peer review of "DRL1, Encoding A NAC Transcription Factor, Is Involved in Leaf Senescence in Grapevine"

_ijms, 2019, doi:10.3390/ijms20112678_

Round 1

Reviewer 1 Report

Submitted manuscript entitled DRL1encoding an NAC transcription factor, is involved in leaf senescence in grapevine presents results on the role of DRL1 and  NAC transcription factor on leaf senescence in selected Vitis vinifera ‘Yatomi Rose’ cultivar.

The scientific goal and hypothesis should be clearly specified in Abstract and Introduction. According to the authors, aging of the leaves is a significant problem in obtaining high quality grapes. However, it is not clear how the results of the research carried out would solve this problem. The discussion is more focused on tobacco and results of other research than on grapevine. Moreover, the text is chaotic and hard to following. Before resubmission the manuscript must be significantly improved (aim of study, hypothesis, discussion).

Specific comments:

L. 34. Citation is needed.

L. 39. Arabidopsis thaliana - Arabidopsis thaliana (L.) Heynh.) – all scientific names first time used in the text should be given with author abbreviation. In all next time shorter form should be used, eg. A. thaliana. Objection also applies to other names used in the text. Currently, the botanical nomenclature is used inconsistently and chaotically.

L. 59-61. ‘Leaf senescence seriously affects photosynthesis and nutrient assimilation, and subsequently influences the yield and quality of grapes.’ - If it is, citation is needed. Moreover, it should be discussed in detail basis of own results.

L. 62. ‘NAC genes in terms of their role in leaf senescence is not available’ -> NAC genes in terms of their role in leaf senescence is unknown.

L. 62-64. ‘In our previous study, a NAC transcription factor gene DRL1 (GenBank No. XP-002281816) was identified from grapevine which was associated with dwarf plant and rolled leaf phenotypes [27].’ – to Discussion.

L. 64-66. In  the present study, we report that DRL1 plays an important regulatory role in ABA-mediated leaf 65 senescence and act as a negative regulator of senescence. – to Results.

L. 98. ‘transgenic plans’ -> transgenic plants

L. 184-185. ‘These results indicated that ABAs play an important role in leaf senescence.’ - This has been confirmed many times in many other studies.

L. 262. Yatomi Rose ->‘Yatomi Rose’

Author Response

Dear editor and reviewers:

Thanks for your advice. According to your comments, we revised the manuscript and explain point-by-point the details of the revisions.

If you have any questions, please contact us. We look forward to hearing from you soon.

The following is our responses to the reviewer’s comments. 

Thank you and best regards.
        Yours sincerely,

                                                                                       Bo Li     ([email protected])

Ziguo Zhu ([email protected])

Reviewer 2 Report

The manuscript by Zhu describes the function of DRL1, a NAC transcription factor, in leaf senescence and hormone regulation in grapevine. Combining the results of gene expression profiles, overexpression transgenic lines, hormone treatments and assessments, and transcriptome analysis, the authors indicated that DRL1 to be one among the negative regulator of leaf senescence by regulating ABA synthesis. The work is original and interesting. The manuscript is globally clear and solid. I have several specific comments for this work:

1. In figure 1, the authors provided the dynamic expression patterns of DRL1 during the leaf senescence. I am also wondering the tissue-specific expression patterns of DRL1. I encourage the authors to examine the expression of DRL1 across different tissues (e.g. flower and root).

2. Line 178, the authors examined the expression change of ABA biosynthesis genes between WT and transgenic lines. I suggest authors to check the genes in ABA signaling pathway, like PYL, PP2C and SnRK2, as well.

3. Line 268, the authors mentioned that leaf samples were collected at “different time points”. I can’t find these time points information in the results. Because of the circadian effects on the NAC family genes, it’s important to clarify the timing information of sample collection.

4. Line 300, "transcriptome analysis" section, bioinformatic pipeline should be clarified, by showing the mapping, filtering and annotation. Mean quality and total amount of reads per sample should be provided, as a Supplementary Table. Raw sequencing data should be uploaded on the public database with accession number.

5. Not a suggestion – just a question: What is the homolog of DRL1 in Arabidopsis and rice?

Author Response

(The authors gave the same response as above.)

Reviewer 3 Report

Dear Editor,

Thank you for considering me as a reviewer for the article, “DRL1encoding an NAC transcription factor, is involved in leaf senescence in grapevine” publication in your esteemed journal IJMS. I have provided my comment as follows.

Broad comments:

The manuscript by Zhu et al, aim is to study function of DRL1 gene (NAC transcription factor) in Grapevine as a negative regulator of leaf senescence. The main contribution of the paper is to report that DRL1 gene is involved in ABA-induced leaf senescence by their experiment and observation. Additionally; GUS staining, chlorophyll/hormone measurement, expression analysis and transactivation activity analysis were performed to make a point. I would recommended that author should include all the raw data, statistical methodology and analysis data generated from each experiment in supplementary material for readers. The following points may be considered while revising the articles.

 Specific comments:

1.      Line 21: “salicylic acid” As per Fig. 4E, it shows significant difference.

2.      Line 29: “Introduction” Please provide more background information about NAC protein family and Leaf senescence.

3.      Line 62: Could you please share what other information or study has been done.  

4.      Line 68: Fig. 1 (b) Describe bar.

5.      Line 87: Please insert graph for displaying ratio of yellow leaf to all leaves per transgenic line.

6.      Line 89: Figure 2; please write X-axis legend for all graphs and “n” meaning.

7.      Line 91: Ion leakage should be label “e” and Fv/Fm ratio should be label “d”.

8.      Line 92: Full form of “SD”.

9.      Line 93: Could you please provide reference for Turkey’s test.

10.  Line 105: Section 2.3; could you please provide heat map for all Regulatory genes (up & down regulated genes) and heat map showing gene associated with Chlorophyll and ABA will be helpful for readers.

11.  Line 107 & 108 and Fig 3a: “DEG” instead of “DAG”.

12.  Line 126 to 129: Please provide validation result for RNA-seq analysis by analyzing expression pattern of top up and down regulated genes from total DEG list.

13.  Line 132: Fig 4a, in IBA is there a significant difference between WT and transgenic.

14.  Line 144: Fig 5; could you please mention data and figure for Day 5 & 9 in Fig 5c &d. Could you please also mention data and figure for Day 7 & 9 in Fig 5e & for Day 5 & 7 in Fig 5f.

15.  Line 186 (Discussion): Please describe reason for using tobacco plant, it will be helpful for readers.

16.  Line 201: Is it possible for you to provide reference for few more studies?

17.  Line 275: please provide detail of probe used in qPCR and mention that Actin was used as a control for normalization.

18.  Line 293: Please provide data for validation of tobacco transgenes and how many transgenic line was used in the study.

19.  Line 295 & 300: Please mention quantity of leaf tissue for extraction and day-light exposure time. Even though you have provided reference but if you provide brief method description it will be useful.

20.  Line 300: Please provide detail description about Transcriptome data analysis.

21.  Line 315: suggestion “Acknowledgments and funding” instead of “Acknowledgments”.

22.  Please provide all the raw data including calculated statistical analysis data into supplementary section after line 322.

23.  Have you tried western blot on OE transgenic lines?

Author Response

(The authors gave the same response as above.)

Round 2

Reviewer 1 Report

I’m satisfied in author response and current version of manuscript entitled DRL1encoding an NAC transcription factor, is involved in leaf senescence in grapevine. This work will be suitable for publishing in IJMS after some technical corrections in botanical nomenclature.

L. 70. Botrytis cinerea – please add author abbreviation

L. 71. Hyaloperonospora arabidopsidis - please add author abbreviation

L. 71. V. Amurensis ->Vitis amurensis, please add author abbreviation